# Betting on Non-Invasive Brain Stimulation to Treat Gambling Disorder: A Systematic Review and Meta-Analysis

**DOI:** 10.3390/brainsci13040698

**Published:** 2023-04-21

**Authors:** Lilia Del Mauro, Alessandra Vergallito, Gaia Gattavara, Lucrezia Juris, Alessia Gallucci, Anna Vedani, Laura Cappelletti, Pietro Maria Farneti, Leonor J. Romero Lauro

**Affiliations:** 1Department of Psychology, University of Milano-Bicocca, 20126 Milano, Italy; 2Fondazione Eris Onlus, 20134 Milano, Italy; 3Department of Psychology & Neuromi, University of Milano-Bicocca, 20126 Milano, Italy; 4Ph.D. Program in Neuroscience, School of Medicine and Surgery, University of Milano-Bicocca, 20900 Monza, Italy

**Keywords:** gambling disorder, craving, tDCS, rTMS, non-invasive brain stimulation

## Abstract

Gambling disorder (GD) is a behavioral addiction that severely impacts individuals’ functioning, leading to high socioeconomic costs. Non-invasive brain stimulation (NiBS) has received attention for treating psychiatric and neurological conditions in recent decades, but there is no recommendation for its use for GD. Therefore, this study aimed to systematically review and analyze the available literature to determine the effectiveness of NiBS in treating GD. Following the PRISMA guidelines, we screened four electronic databases up to July 2022 and selected relevant English-written original articles. We included ten papers in the systematic review and seven in the meta-analysis. As only two studies employed a sham-controlled design, the pre–post standardized mean change (SMCC) was computed as effect size only for real stimulation. The results showed a significant effect of NiBS in reducing craving scores (SMCC = −0.69; 95% CI = [−1.2, −0.2], *p* = 0.010). Moreover, considering the GD’s frequent comorbidity with mood disorders, we ran an exploratory analysis of the effects of NiBS on depressive symptoms, which showed significant decreases in post-treatment scores (SMCC = −0.71; 95% CI = [−1.1, −0.3], *p* < 0.001). These results provide initial evidence for developing NiBS as a feasible therapy for GD symptoms but further comprehensive research is needed to validate these findings. The limitations of the available literature are critically discussed.

## 1. Introduction 

Gambling can be defined as a behavior that requires wagering something valuable at risk in hopes of winning something of higher value [1]. While ‘pathological gambling’ was previously classified under the section ‘impulse control disorders not elsewhere classified’, increasing behavioral and neurobiological evidence suggests that the disorder has more in common with substance-use disorders (SUDs). Therefore, the fifth edition of the Diagnostic and Statistical Manual (DSM-5) classified gambling disorder (GD) among substance-related and addictive disorders [2]. Similarly, the International Classification of Diseases, eleventh edition (ICD-11) [3] (https://icd.who.int/en accessed on 25 February 2023) moved GD to conditions due to addictive behaviors. According to both systems, GD is characterized by persistent and recurrent maladaptive gambling behavior causing clinically significant distress and impairment in individuals’ areas of functioning, such as personal, social, educational, occupational, and socio-economic.

The clinical features of GD can be clustered into (i) loss of control, concerning the increasing amount of money used for gambling and unsuccessful efforts in controlling, reducing, or stopping gambling; (ii) gambling urgency or withdrawal, including gambling when feeling distressed and showing restless or irritable mood when trying to reduce or stop gambling; (iii) neglect of other areas in life, as indicated by the lies to cover gambling, relying on others to provide the money, and loss of significant relationships, employment, and educational or career opportunities [4].

At a clinical level, GD has high rates of chronicity [5], with patients typically reporting a low quality of life, poor health conditions, and increasing suicide rates [6]. GD often leads to severe maladaptive consequences for the individuals suffering from the disorder, family members, and social surroundings. According to a previous systematic review [7], GD’s prevalence in the general population varied between 0.12% and 5.8%, with differences according to the country and the specific screening instrument employed to estimate gambling behavior rates. Notably, GD’s prevalence increases when considering clinical populations, unveiling the high comorbidity with other psychiatric diseases, especially mood and anxiety disorders, substance abuse and dependence [8,9], or cognitive dysfunctions [10,11]. Indeed, 96% of individuals with GD suffer at least from another psychiatric disorder, and 64% satisfy the criteria for three or more psychiatric conditions [8], thus complicating the individuals’ clinical picture, treatment adherence, and therapeutic outcome.

Although the behavioral and neurobiological overlap between SUDs and GD is well documented and has contributed to updating diagnosis manuals, the DSM-5 and ICD-11 criteria for GD do not include an item directly assessing the presence of craving, differently from the ones for SUDs. Craving can be defined as a pressing, urgent, and irresistible desire to give in to addictive behavior, which is typically followed by compulsive research into the target action (e.g., alcohol consumption). Moreover, craving is commonly associated with the illusory expectation of positive reinforcement and the relief of negative states resulting from implementing the behavior of interest. Craving can be triggered by specific cues that can be external (e.g., visual stimuli associated with the habits of addiction) or internal (e.g., emotional states), or can occur independently. Craving predicts relapse, even after long periods of abstinence, thus being crucial in maintaining the dependence [12]. Growing evidence actually suggests that craving plays a role in GD, as in other behavioral addictions [13], with GD patients often showing a loss of control over gambling behavior to retrieve significant losses or attain higher winnings. According to a model developed by Brand and colleagues [14] for internet-use disorder and updated for addictive behaviors (Interaction of Person–Affect–Cognition–Execution, I-PACE), gambling behaviors would be due to the interactions between predisposing factors, such as affective and cognitive reactions to trigger stimuli, impaired executive functions (primarily inhibitory control), and decision-making processes. From this perspective, the associations between craving/cue reactivity to trigger stimuli and diminished inhibitory control would contribute to GD’s development and maintenance.

A recent systematic review [15] investigated the role of craving in adults reporting different levels of gambling (from non-clinical gamblers to individuals satisfying the criteria for diagnosis). The authors included 62 papers highlighting the relevance of craving in GD. Craving predicted the occurrence of gambling episodes [16], had a positive correlation with GD severity, negative urgency and emotions, and a negative correlation with positive emotional states [9,17,18].

The etiology of GD is complex and needs to be wholly clarified. It includes both genetic and environmental factors [19]. Similarly, at a neurobiological level, the brain mechanisms underlying GD are far from being fully understood. Previous evidence shows structural and functional abnormalities partially overlapping the ones described in SUDs [20,21,22,23], including the frontostriatal and limbic networks, the orbitofrontal cortex, the anterior cingulate cortex, the insula, the hippocampus, and the amygdala [19]. Recent findings highlighted the role of the striatum in reward processing and stimulus–outcome association [24]. It has been observed that, when faced with gambling cues and monetary incentives, or the action of gambling, individuals with GD and SUDs exhibited reduced striatal activation during reward anticipation. Differently, the prefrontal networks play a pivotal role in decision-making, particularly regarding decisions influenced by rewards [23]. Extensive evidence has highlighted abnormalities in the frontostriatal areas when GD patients are presented with gambling-related cues [25], during experimental gambling tasks [26], or when required to inhibit a response (i.e., during a Stroop task) [27]. The exposure to gambling-related cues in the GD sample also led to more pronounced activity in the insula and anterior cingulate cortex [28]. In contrast, diminished volume in the hippocampus and amygdala compared with healthy controls was observed [29]. The malfunctioning of the reward, motivational, and cognitive control circuits in GD individuals accounts for the clinical features of the disease, such as increased sensitivity to reward, executive dysfunctions, impulsive decisions, stress dysregulation, and social–emotional problems.

Considering the feasibility of treating GD, the first issue is the low percentage (~10%) [30,31,32] of individuals seeking professional support or participating in self-help groups, such as Gamblers Anonymous. Even when seeking treatment, patients typically exhibit high dropout rates before treatment completion [33,34], scarce commitment, and frequent relapses [35,36]. For individuals completing treatment, the estimated rates related to the success of therapy are inconsistent [37], probably due to the heterogeneity in the outcome measures considered to evaluate treatment efficacy in this population [38,39]. Besides the critical points related to GD patients’ compliance, no consensus has been reached on a gold-standard treatment, although a broad panel of options is available considering psychological and pharmacological interventions [19,40]. Traditionally, psychological interventions, such as individual psychotherapy, are the preferred treatment option, although they have shown only short-term efficacy.

Innovative and more effective treatment protocols are thus required. In this context, non-invasive brain stimulation techniques (NiBS) have received attention based on previous results in treating other psychopathological conditions, such as Major Depressive Disorder (MDD), anxiety, obsessive–compulsive disorder, schizophrenia, and SUDs [41,42,43,44,45].

Among NiBS, the two techniques mainly used are transcranial magnetic stimulation (TMS) and transcranial direct current stimulation (tDCS). TMS is a neurostimulation technique that delivers a strong and short magnetic pulse over the patient’s head. The pulse induces neuronal firing by suprathreshold neuronal membrane depolarization. In the clinical field, TMS is typically applied using repetitive (rTMS) protocols and is able to generate neuroplastic effects through long-term potentiation (LTP) and depression (LTD)-like changes [46,47]. Protocols are typically delivered to inhibit (≤1 Hz and continuous theta-burst stimulation, cTBS) or excite (>5 Hz and intermittent theta-burst stimulation, iTBS) the stimulated brain networks. The focality and depth of the stimulation depend on the coil’s geometry and size. While traditional coils (circular, figure of eight/butterfly) allow for stimulating superficial cortical regions, H-coils have led to a non-focal stimulation reaching brain areas distant from the cortical surface up to 4–5 cm [48]. Differently, tDCS is a neuromodulatory technique that delivers a weak constant current (typically 1–2 mA) through two electrodes, an anode and a cathode, placed over the scalp. Unlike TMS, the intensity of tDCS is not strong enough to elicit action potentials, but influences potential membrane excitability by depolarizing (anodal) or hyperpolarizing it (cathodal) [49,50].

Regarding the treatment of SUDs, the rationale for using NiBS relies upon preclinical studies, which have highlighted the relevance of prefrontal cortex (PFC) dysfunctions in maintaining addictive behaviors. Interestingly, Chen and colleagues [51] employed a compulsive cocaine-seeking rat model, showing that the drug self-administration persisted despite the delivery of nociceptive shocks. Moreover, the authors reported reciprocal results considering the role played by the rat’s prelimbic cortex (PLC). Indeed, prolonged exposure to drug self-administration reduced PLC excitability; similarly, in vivo PLC optogenetic stimulation significantly prevented drug-seeking behaviors. These findings drove the rationale for applying NiBS to the PFC—especially the dorsolateral prefrontal cortex (DLPFC)—to treat SUDs. Indeed, the human PFC is considered the homologous human region of the rat PLC [52,53], and the DLPFC, in particular, is considered its functionally equivalent region [54,55]. Crucially, the human DLPFC plays a central role in decision-making processes, executive control, such as inhibitory and attentional abilities, and cue-induced craving [56,57]. In line with these considerations, a previous review highlighted that stimulating the DLPFC improved gambling-related decision processes, such as delay discounting or loss-chasing [58].

Several reviews and meta-analyses investigated the feasibility of applying NiBS to SUDs [51,59,60,61,62,63,64,65,66]. Globally, promising results have been found, although they have been hampered by the high variability across studies in the stimulation protocols, sample characteristics, outcome measures, and research design. Studies typically focused on monitoring craving in the short or medium term through subjective measures, often lacking in tracking drug consumption and relapse frequency. For instance, a recent meta-analysis highlighted a significant effect of high-frequency rTMS over the left DLPFC in reducing craving in substance-addictive behaviors directly based on the dopamine system, i.e., cocaine, amphetamine, or methamphetamine users [67]. Contrasting results have been reported for alcohol addiction. A meta-analysis by Kim and colleagues [68] highlighted a small, but significant, effect of bilateral stimulation of DLPFC, with the anode targeting the right and the cathode the left DLPFC. In contrast, a previous meta-analysis investigating tDCS/rTMS effects did not highlight evidence of efficacy [63].

Independent panel experts periodically revise the literature, providing recommendations for or against stimulation across psychiatric and medical conditions. High-frequency rTMS delivered over the left DLPFC received a level C “possible efficacy” for nicotine craving and consumption [69]. Considering tDCS, a level B “probable efficacy” was recognized for bihemispheric DLPFC stimulation (anodal right–cathodal left) for alcohol addiction [70]. No recommendations are currently available for GD due to the limited number of studies targeting the disease.

To our knowledge, only a previous systematic review focused on the use of NiBS in GD [71] without providing a quantitative measure of studies’ effects. To fill this gap, in the present work, we aimed to systematically update and quantitatively analyze the available data on the topic to provide evidence on the literature’s state-of-the-art, limitations, and future directions. Due to the relevance of craving symptomatology within GD, we focused on craving scores as the primary outcome measure. In addition, considering the frequent comorbidity of GD with mood disorders, we addressed depressive symptoms as secondary outcome measures.

## 2. Methods

### 2.1. Literature Search

Following the Preferred Reporting Items for Systematic Reviews and Meta-Analyses (PRISMA) guidelines [72], we screened PubMed, EMBASE, Web of Science, and Scopus to select papers published before 3 July 2022. We built our search queries by combining keywords related to NiBS (using both the acronyms and the full words) with gambling-related keywords. We used the following keywords: “brain stimulation”, “Non Invasive Brain Stimulation”, “NIBS”, “Transcranial Magnetic Stimulation”, “TMS”, “repetitive Transcranial Magnetic Stimulation”, “rTMS”, “Theta Burst Stimulation”, “TBS”, “deep Transcranial Magnetic Stimulation”, “dTMS”, “deepTMS”, “transcranial Direct Current Stimulation”, “tDCS”, “transcranial Electrical Stimulation”, “tES”, “transcranial Alternating Current Stimulation”, “tACS”, “transcranial Current Stimulation”, “transcranial Current Stimulation”, “tCS”, “pathological gambling”, “gamblers”, “gambling”, “gambling disorder”, and “GD” (see Appendix A for details of the search strategies). Papers were eligible if they: (a) were written in English, (b) involved humans, (c) were original research published in scientific journals, (d) involved patients with a diagnosis of gambling disorder or pathological gambling, and (e) used non-invasive brain stimulation techniques for treatment purposes. Therefore, we excluded non-English written papers, articles on animal samples, conference proceedings and abstracts, theses, book chapters, systematic and narrative reviews, meta-analyses, single case studies, research including healthy participants or individuals defined as “at-risk gamblers”, and those applying NiBS without pursuing the objective of treating GD symptoms.

### 2.2. Records Screening and Data Extraction

The screening process was run using Rayyan (https://rayyan.qcri.org/ accessed on 7 July 2022), a web and mobile systematic reviews manager [73]. Three blinded researchers (L.D.M., G.G., and L.J.) removed duplicates and independently screened the titles and abstracts of records retrieved from the search databases. Considering the eligibility criteria, we used specific labels to classify the documents as “included”, “excluded”, or “maybe”. The latter was chosen when the title and abstract lacked sufficient information to be included or excluded certainly. After that, records in the “included” and “maybe” categories were considered for the second screening stage according to the articles’ full texts, which were independently analyzed and screened by three blinded researchers (L.D.M., G.G., and L.J.) to select eligible papers. In this screening stage, the corresponding authors were contacted when the records’ full texts were unavailable. In both screening phases, conflicts were solved by consensus or involving a fourth author (A.G.). Three structured tables (see the Results section for details) were used by three researchers (L.D.M., G.G., and L.J.) to extract data from the included studies. The tables were checked for consistency and accuracy by another author (A.G.), and discrepancies were solved by consensus.

### 2.3. Studies Quality Assessment

The quality of the eligible studies was independently assessed by three blinded researchers (L.D.M., G.G., and L.J.) based on the items of the Revised Cochrane Risk-of-Bias 2 Tool for Randomized Trials (RoB 2) [74] and the National Institutes of Health (NIH) quality assessment tool for before–after (Pre–Post) studies with no control group (https://www.nhlbi.nih.gov/health-topics/study-quality-assessment-tools accessed on 16 March 2023). The former tool evaluates the following bias domains: (a) randomization process (“low-risk” judgment: use of a clear strategy—e.g., random sequence—to randomize participants; concealment of groups’ allocation until participants were enrolled and assigned to interventions; and involvement of balanced study groups based on baseline differences); (b) intervention assignment (“low-risk” judgment: studies using double-blind approach); (c) missing data (“low-risk” judgment: studies with outcomes’ data available for all, or nearly all, participants); (d) outcomes’ measurement (e.g., appropriateness of the method of measuring outcomes and the influence of blinding on the outcomes’ measurement); and (e) selection of the reported results (“low-risk” judgment: declaration of a pre-specified analysis plan and reporting of results that are not intentionally selected among the other estimates based on their magnitude or statistical significance). Each domain was rated as “Yes” or “Probably Yes”, “No” or “Probably No”, or “No Information” when the papers lacked sufficient information to provide a “Yes” or “Probably Yes”, or a “No” or “Probably No” judgment. The NIH tool is made up of 12 items assessing the clarity of the study question and participants’ eligibility/inclusion criteria, the representativeness of the sample for the clinical population of interest, the sample size’s appropriateness, the description clarity of the delivered intervention, the reliability of the outcome measures employed, the blindness of the people administering the outcome measures for participants’ intervention, the potential influence of the loss of assessments after baseline on the results, the appropriateness of the statistical methods, and the presence of follow-ups. Each item was evaluated with a “Yes”, “No”, or “Other” response when the specific item could not be determined or applied to the study, or when the paper lacked sufficient information to provide a “Yes” or “No” response. Finally, an overall quality judgment equal to “Good”, “Fair”, or “Poor” was assigned based on the ratings assigned to each item. Conflicts in the quality assessment were solved by consensus of the three researchers or by involving another author (A.G.) when needed.

### 2.4. Quantitative Analysis

We extracted relevant information for each of the included studies. We collected information considering the NiBS protocol applied (technique, number of sessions, and target regions) and participants’ characteristics (sample size, age, and gender). As only 2 articles included a sham condition in crossover designs, the quantitative analysis was run on pre–post scores only for the active stimulation condition. As the primary outcome measure, we extracted pre- and post-treatment means and standard deviations of validated instruments—when available—and visual analog scale (VAS) scores assessing craving (notes concerning the included measures are detailed in Appendix A). Moreover, considering the comorbidity with mood disorders, we included an exploratory meta-analysis on depressive symptoms (4 studies included). When the information reported in the main text, tables, or Appendix A was insufficient, we tried to contact the paper’s authors to obtain the missing data.

For each included study, we calculated the sampling variance and the standardized mean change (SMCC) using change score standardization [75], computed by the “escalc” function of the “metafor” package for R (version 3.4.3) [76,77]. SMCC is typically used to assess the amount of change within a group, for example, before and after treatment, as in our case. The correlation between pre- and post-measurement variances was set at 0.5, as suggested by Follmann and colleagues [78]. We ran sensitivity analyses establishing lower (0.25) and higher (0.75) correlations [79,80] to ensure that this arbitrary choice did not influence our results. None of these assumptions caused a meaningful change in the results. For the sake of clarity, the results of these analyses are reported in Appendix A.

Considering the included articles, only one [81] had sufficient information to calculate two effect sizes for the VAS. Indeed, the authors delivered both 10 Hz rTMS and cTBS stimulations in different sessions. Considering the effect sizes coming from the same article as statistically independent would violate the independence assumption of traditional meta-analyses and create bias in the statistical findings [82]. However, given that this issue was present in only one of the studies, we ran a multi-level (three-level) model using the ‘rma.mv’ function of the ‘metafor’ package, clustering the effect size at the study level. Then, we compared it with a simpler model (not including the three-level model) using the ‘anova’ function [83]. Given that no difference was detected between the two models (*p* = 0.127), we used the simpler one and applied a random-effects model with the ‘rma’ function of the ‘metaphor’ R package. We chose a random-effects model because it is suitable for dealing with heterogeneity due to sampling error and variance in studies’ effect sizes [84].

In line with suggestions from the literature [83], we provided several measures to characterize the heterogeneity of our data. We reported the variation due to the sampling error (Q statistic), the percentage of variation between studies not due to the sampling error (I^2^ statistics) [85], and the prediction intervals (PIs) [86] that provide a range into which we could expect the effects of future studies effects to fall based on the available data.

The influence function ‘inf’ implemented in the metafor package was applied to detect outliers and influential cases. In line with Viechbauer and Cheung’s recommendations [77], when outliers and influential cases were detected in the dataset, we refitted the model after removing those cases. This procedure is helpful for verifying that the elimination of extreme data points does not impact the analysis results, and that the findings are robust and do not hinge on a few unusual effect sizes. Crucially, studies identified as potential outliers were scrutinized for their contents to understand whether patterns acting as potential moderators could explain their effect size magnitude. Considering potentially interesting moderators of the NiBS effect, several features can impact treatment outcomes, including the number of sessions, the type of stimulation, and the interaction between brain areas. However, considering the limited number of studies in the present meta-analysis, we decided to run only a meta-regression, including the number of sessions as a continuous moderator.

Finally, we ran an exploratory meta-analysis on the secondary outcome measure, measuring the effects of NiBS on the modulation of depressive symptoms.

## 3. Systematic Review Results

### 3.1. Studies Selection

We retrieved a total of 5037 records from the four screened databases. A group of 702 documents was removed as duplicates, while 4335 were screened based on the articles’ titles and abstracts. Among these, 4315 were excluded as they did not meet the eligibility criteria. After examining the full texts of the remaining 20 papers, 9 studies were included in the qualitative synthesis. Among the 11 studies excluded, 2 papers involved no-gambling samples [87,88], 5 were not original research [89,90,91,92,93], and 4 papers used either TMS [94,95] or tDCS [96,97], but not for treatment purposes.

Moreover, we included the data from a study by our group [98] that is currently in pre-print on OSF (https://osf.io/ufd93/ accessed on 10 March 2023). The study aimed to assess the feasibility of administering tDCS to GD patients to reduce craving and gambling-related symptoms. Most of the eligible studies considered clinically validated questionnaires to test the effects of NiBS on craving.

Only one paper [99] administered behavioral tasks (i.e., the Iowa Gambling Task, IGT, and the Wisconsin Card Sorting Test, WCST) to GD patients to evaluate the efficacy of NiBS in enhancing cognitive flexibility and decision-making abilities. Therefore, to keep homogeneity among the studies included in the quantitative analysis, we included this paper in the systematic review, but not in the meta-analysis. Other two studies were included in the systematic review, but not in the meta-analysis [100,101]. The paper by Martinotti and colleagues [100] included a sample of 34 SUDs, with only 4 participants (2 treated with active stimulation and 2 with sham) having a GD diagnosis. The study by Sauvaget et al. [101] did not provide sufficient information to be included. The screening process is summarized in Figure 1.

### 3.2. Quality Assessment

The results of the quality assessment are reported in Table 1 and Table 2. Five papers [81,99,100,101,102] were evaluated through the Rob-2, as they were randomized trials employing a sham-controlled design. The quality assessment resulted in an overall risk of bias judgment of “some concern” due to the lack of detailed information on the randomization procedure [81,99,100,101] or the absence of a pre-specified analysis plan [81,101,102].

The five papers without a control group [98,103,104,105,106] were evaluated using the NIH tool (see Appendix A for details). Two papers [98,104] received an overall judgment of “Good”, whereas the other two works [103,106] were rated “Fair”. This rating arose mainly due to the small sample sizes of the studies, and because the treated sample was not evaluated as representative enough of GD patients. Indeed, in some studies authors excluded patients with SUDs and mood disorders, that are frequently associated with GD,. Finally, the study by Rosenberg and colleagues [105] received an overall judgment of “Poor” due to the small sample size, by the lack of clarity in the description of the participants’ inclusion criteria, and by the fact that one patient was treated twice.

### 3.3. Participants’ Characteristics

We included 10 studies in the systematic review, resulting in 159 participants, with a high prevalence of males (N = 140). Patients ranged from 18 to 70 years of age (mean age = 41.8, SD = 4.50), and secondary school was the most common educational level. All participants received a diagnosis of ‘pathological gambling’ according to DSM-IV [81,101] and DSM-IV-TR [105], or ‘GD’ according to DSM-5 criteria [98,99,102,103,104,106]. One paper [100] did not specify the criteria for which participants were diagnosed with a GD (i.e., DSM-IV, DSM-5, or ICD-11). Considering the exclusion criteria, most studies did not include participants with other SUDs. In one study [98], 3 out of 18 patients were polyabusers, and another [103] included patients with GD in comorbidity with cocaine use disorder. Nine out of ten studies included treatment-seeking patients, while Zack et al. [81] recruited non-treatment-seeking individuals. However, only two studies [98,104] clarified the associated therapies, which included psychological, educational, and psychosocial interventions. Overall, the patients were drug- and medication-free or under a stable pharmacological regimen, although stability was defined differently across studies (e.g., 1 month or 6 months).

Most of the studies excluded participants with psychiatric comorbidities, such as schizophrenia or other psychosis [98,99,100,102,103,104] and mood disorders [99,100,106]. Participants should satisfy the safety guidelines for NiBS administration [107,108,109]. Details of the studies’ additional inclusion and exclusion criteria are reported in Appendix A. Table 3 reports the descriptive details of the included samples.

### 3.4. Stimulation Protocols

Of the included studies, seven applied TMS protocols [81,101,102,103,104,105,106] and three used tDCS [98,99,100]. The studies were heterogeneous regarding stimulation parameters and the number/frequency of delivered sessions. In the following paragraphs, the studies will be discussed according to the type of NiBS employed, namely tDCS (see Table 4 for details on stimulation protocols) and TMS (protocols details in Table 5).

### 3.5. TDCS Studies

The studies employing tDCS converged in stimulating the DLPFC bilaterally, with the anode placed over the right DLPFC and the cathode over the left DLPFC. The duration was also comparable in the three studies, where tDCS was delivered for 20 min. The stimulation intensity and electrode size varied among studies: Martinotti et al. [100] performed stimulation at 1.5 mA using 5 × 5 electrodes (current density = 0.06), while Del Mauro and colleagues [98] and Soyata et al. [99] performed stimulation at 2 mA using 5 × 5 (current density = 0.08) and 5 × 7 (current density = 0.057) electrodes, respectively. Two studies ran a randomized sham-controlled parallel design with double [100] or triple [99] blinding, whereas Del Mauro and colleagues [98] administered only active tDCS in an open-label design. The number and frequency of sessions varied in the three studies. The tDCS protocol in Soyata et al. [99] resulted in three every-other-day tDCS sessions (active or sham). In contrast, Martinotti et al. [100] delivered one daily tDCS session (active or sham) for 5 consecutive days. Finally, Del Mauro and colleagues [98] administered six active tDCS sessions over 2 weeks (three tDCS sessions per week with a washout period of at least 24 h). See Table 4 for details.

**Table 4 brainsci-13-00698-t004:** Detailed information about tDCS studies protocols and results.

References	Stimulation Site	Stimulation Protocol	Craving Measures Included	Other Measures (Clinical/Behavioral)	Follow-Up	Study Design	Blinding	Results
Del Mauro et al. [98]	Left and right DLPFC	Anode right DLPFC and cathode left DLPFC. Intensity 2 mA, 20 min, electrodes 5 × 5, three sessions per week for two weeks.	MATE	VAS; SCQ; BDI; SCL-90 (GSI index); WHOQOL-B; and BIS-11	At 3 and 6 months after treatment	Open label	NR	Improvements in craving scores, desire to use substances, mood, impulsivity, and quality of life
Martinotti et al. * [100]	Left and right DLPFC	Anode right DLPFC, cathode left DLPFC; 1.5 mA, 20 min, electrodes 5 × 5, 5 consecutive days (1 tDCS session per day)	VAS	BIS-11; gambling behaviors/substance consumption Timeline Follow-Back; HAM-D; HAM-A; and Y-MRS	None	Randomized sham-controlled	Double-blind	Overall reductions in anxiety, mood, impulsivity, and craving scores. In the latter, larger effects in the real condition
Soyata et al. * [99]	Left and right DLPFC	Anode right DLPFC, cathode left DLPFC; 2 mA, 20 min, electrodes 5 × 7, 3 every-other-day sessions (active or sham)	NR	SOGS, PGSI and BIS-11 (at baseline); IGT; and WCST	None	Randomized sham-controlled	Triple-blind	Improvements in cognitive flexibility and decision-making processes in the real tDCS group

Barratt Impulsiveness Scale (BIS-11); Beck Depression Inventory (BDI); dorsolateral prefrontal cortex (DLPFC); global severity index (GSI); Hamilton Anxiety-Rating Scale (HAM-A); Hamilton Depression-Rating Scale (HAM-D); Iowa Gambling Test (IGT); Measurements in the Addictions for Triage and Evaluation (MATE); not reported (NR); Pathological Gambling Severity Index (PGSI); Symptoms Checklist 90 (SCL-90); South Oaks Gambling Screen (SOGS); Substance Craving Questionnaire (SCQ); Visual Analog Scale (VAS); Wisconsin Card Sorting Test (WCST); World Health Organization Quality of Life–Brief (WHOQOL-B); Young Mania Rating Scale (Y-MRS). * The studies were included in the systematic review and excluded from the meta-analysis.

### 3.6. TMS Studies

The majority of studies applied an excitatory high-frequency rTMS at 10 [81,102] or 15 Hz [103,104], targeting either the left DLPFC [102,103,104] or the medial prefrontal cortex (mPFC) [81]. Two studies applied a low-frequency inhibitory protocol, delivering rTMS at 1 Hz [101,105], with Rosenberg and colleagues [105] applying deep rTMS to stimulate the left DLPFC and Sauvaget and colleagues [101] targeting the right DLPFC. Finally, two studies employed cTBS protocols [81,106] to stimulate the right DLPFC [81] or the pre-supplementary motor area (pre-SMA) bilaterally [106]. Among most of the papers employing the traditional “figure-of-eight coil” [81,100,101,102,106], Rosenberg et al. [105] used the H1 coil. Concerning the design, three studies were open-label studies [104,105,106], whereas three were double-blind sham-controlled crossover studies [81,101,102]. Finally, one paper [103] reported a case series including seven treatment-seeking patients. The studies were quite heterogeneous regarding the number and frequency of delivered sessions. Two articles [103,104] employed an intensive phase consisting of a high number of stimulation sessions delivered over a brief period (i.e., 10 TMS sessions for 5 consecutive days [103], and twice daily for 5 days for 2 weeks [104]), followed by a maintenance phase in which follow-up stimulations were delivered over a longer amount of time (i.e., twice daily once a week for 8 weeks [103], and twice daily once a week for 12 weeks [104]). The number of delivered sessions in other studies ranged from 2 to 15, with different washout periods. See Table 5 for details.

**Table 5 brainsci-13-00698-t005:** Detailed information about TMS studies’ protocols and results.

References	Stimulation Site	Stimulation Protocol	Craving Measures Included	Other Measures (Clinical/Behavioral)	Follow-Up	Study Design	Blinding	Results
Cardullo et al. [103]	Left DLPFC	Twice daily for 5 consecutive days and twice daily once a week over 8 weeks; 15 Hz, 100% of the RMT; 60 impulses per train, 15 s ITI, 40 trains, 13 min duration	G-SAS	CCQ; PSQI; BDI-II; SAS; and SCL-90-R (GSI index)	After 5, 30, and 60 days of treatment	Case series	NR	Gambling severity, craving for cocaine, and negative-affect symptoms improved after treatment and at the follow-ups
Gay et al. [102]	Left DLPFC	Two sessions of active and sham rTMS (1-week washout); 10 Hz, 110% of the RMT, 94 trains, 10 s ITI, 3008 pulses in total	VAS cue-induced	NCs control; NCs desire; and PG-YBOCS	None	Randomized sham-controlled crossover	Double-blind	Improvements in cue-induced craving after the real rTMS. No changes in gambling behavior 7 days after
Pettorruso et al. [104]	Left DLPFC	Twice daily, 5 days a week for 2 weeks (20 sessions) and twice daily once a week for three months (24 sessions); 15 Hz, 100% RMT, 60 pulses per train, 15 s ITI, 40 trains, 2400 pulses, 13 min duration	PG-YBOCS	G-SAS; Gambling behaviors Timeline Follow Back; BDI; and SAS²	2, 4, 8, and 12 weeks	Open label	NR	Improvements in gambling severity and the days spent gambling after the intensive and maintenance phases
Rosenberg et al. [105]	Left DLPFC	15 sessions (1 session/day), 1 Hz, 110% RMT, 10 min duration	VAS	DAGS; Y-BOCS; HDRS; HARS; SOGS; CGI-I; and SAS³	Families’ interviews	Open label	NR	The authors reported no significant effect. Scores seemed to reduce, but no statistical analysis was provided
Salerno et al. [106]	Pre-SMA, bilaterally	10 sessions of cTBS. CTBS consists of bursts of 3 pulses separated by 20 ms (i.e., 50 Hz), with each triplet repeated every 200 ms (i.e., 5 Hz); 80% of RMT, 2 trains of 600 pulses, separated by 1 min, a total of 1200 pulses	PG-YBOCS	GUQ; BIS-11; HAM-A; HAM-D; SDS; CGI; and FTND	After the 10 sessions and after 30 days	Open label	NR	Significant improvement in GD severity and CGI after treatment and at follow-up
Sauvaget et al. * [101]	Right DLPFC	Two sessions of active and sham rTMS (1-week washout); 1 Hz, 120% of RMT, one train, 360 pulses, 6 min duration	VAS	GACS; heart rate; and blood pressure	None	Randomized sham-controlled crossover	Double-blind	Improvement in the urge to gamble after treatment. No differences between real vs. sham stimulations
Zack et al. [81]	mPFC, right DLPFC	Three sessions: rTMS, cTBS, and sham (1-week washout). RTMS: mPFC, 80% of AMT, 3 epochs of 15 10-pulse trains, 10 Hz, 10 s ITI, 450 pulses. CTBS: right DLPFC, 80% of AMT, 3 cTBS epochs, 50 Hz, 900 pulses in total	VAS pre and post-TMS	DDT; Stroop task; blood pressure; ARCI; POMS-vigor scale; and VAS pre and post-slot machine game	None	Sham-controlled crossover	Double-blind	Reduction in the desire to gamble after rTMS, but not cTBS.

Active motor threshold (AMT); Addiction Research Center Inventory (ARCI); Barratt Impulsiveness Scale (Bis-11); Beck Depression Inventory (BDI); Beck Depression Inventory-II (BDI-II); Clinical Global Impression Scale (CGI); Clinical Global Impressions—Improvement Scale (CGI-I); Cocaine Craving Questionnaire (CCQ); continuous theta burst stimulation (cTBS); Dannon and Ainhold gambling scale (DAGS); delay discounting task (DDT); dorsolateral prefrontal cortex (DLPFC); Fagerstrom Test for Nicotine Dependence (FTND); Gambling Craving Scale (GACS); Gambling Symptom Assessment Scale (G-SAS); Gambling Urges Questionnaire (GUQ); Global Severity Index (GSI); Hamilton Anxiety-Rating Scale (HAM-A); Hamilton Anxiety Scale (HARS); Hamilton Depression-Rating Scale (HAM-D); Hamilton Depression-Rating Scale (HDRS); Inter-Train-interval (ITI); medial prefrontal cortex (mPFC); not reported (NR); numerical scale (NSc); Pathological Gambling Adaptation of the Yale–Brown Obsessive–Compulsive Scale (PG-YBOCS); Pittsburgh Sleep Quality Index (PSQI); profile of mood states (POMS); repetitive transcranial magnetic stimulation (rTMS); resting motor threshold (RMT); Self-rating Anxiety Scale (SAS); Sheehan Disability Scale (SDS); Symptoms checklist 90–Revised (SCL-90-R); Social Adjustment Scale (SAS³); South Oaks Gambling Screen (SOGS); superior motor area (SMA); transcranial magnetic stimulation (TMS); Visual Analog Scale (VAS); Yale–Brown Obsessive–Compulsive Scale (Y-BOCS); Zung Self-Rating Anxiety Scale (SAS²). * Studies were included in the systematic review, but not in the meta-analysis.

### 3.7. Outcome Measures

Different scales were employed to evaluate the efficacy of NiBS in reducing craving. Most of the studies [81,98,100,101,102,105] used the VAS [110], administered in the form of a 0–10 cm line aimed at ranking patients’ desire to gamble concerning the moment of its presentation. Moreover, Gay and colleagues [102] asked participants to assess, from 0 to 10, their desire to gamble through the “Numerical Scale Desire” (NSc Desire). Other craving measures were validated self-report questionnaires, namely the Gambling Symptom-Assessment Scale (G-SAS) [111] used by [103,104], the Dannon and Ainhold Gambling Scale [112] (DAGS) [105], the Gambling Urge Questionnaire (GUQ) [113] resulting from an adaptation of the Alcohol Urge Questionnaire (AUQ) [114] and used by [106], the Pathological Gambling Yale–Brown Obsessive–Compulsive Scale (PG-YBOCS) [102] employed by [102,104,106], and the Gambling Craving Scale [115] (GACS) administered by [101]. Additionally, Del Mauro and colleagues [98] also employed the Measurements in the Addictions for Triage and Evaluation (MATE)—Q1 Craving form [116], a 5-item scale assessing the perceived disease elicited by the urge to gamble, and the Substance Craving Questionnaire (SCQ) [117], a 47-item questionnaire evaluating different dimensions of craving (i.e., the desire and intention to gamble, the anticipation of positive outcomes derived from the addicted behavior, the anticipation of relief from withdrawal symptoms and dysphoria, and the lack of control related to addicted behaviors). Only one study [103] employed the Cocaine Craving Questionnaire (CCQ) [118], as the authors recruited GD patients with cocaine use disorder comorbidity.

As secondary endpoints, depressive symptoms were assessed through the Beck Depression Inventory (BDI) [98,103,104,119] and the clinician-rated Hamilton Depression Rating Scale (HAM-D) [100,105,106,120].

Only neuropsychological tests (i.e., IGT and WCST) were administered in the study by Soyata and colleagues [99] to assess decision-making and cognitive flexibility abilities.

Three out of ten papers [98,105,106] included follow-up assessments to evaluate treatment outcomes over a medium-term period. Follow-ups were conducted 30 days after the end of the stimulation protocol by Salerno and colleagues [106] and at 3 and 6 months after treatment in the study by Del Mauro et al. [98]. Crucially, Rosenberg and colleagues [105] used family interviews to monitor treatment effectiveness over time without specifying the modality of the interviews, nor when they occurred.

## 4. Meta-Analysis Results

### 4.1. Primary Endpoint: Craving Scores

The meta-analysis was run on eight effect sizes computed from seven papers [81,98,102,103,104,105,106] (total number of participants = 75, mean age = 43.1, SD = 4.2). The aim of the procedure was to compute the effect of NiBS intervention on craving scores. The random-effects model highlighted a significant moderate impact of stimulation symptoms’ reduction after treatment (SMCC = −0.69; 95% CI = [−1.2, −0.2], *p* = 0.010; see Figure 2 for the forest plot).

The Q-statistic, I2, and PIs suggest a high heterogeneity among studies, with values of Q_(7)_ = 24.8 *p* < 0.001, 71.8% CI [0, 96.2], and PIs CI [−2.3, 0.9], respectively. 

Baujat plot inspection (Figure 3) suggested that studies 1 and 6 [103,106] greatly contributed to the heterogeneity of the meta-analysis. The influence analysis confirmed study 1 [103] as an outlier. Therefore, the random-effects model was re-run, excluding this study from the pool. The effect of NiBS on craving reduction remained significant, although the effect slightly decreased (overall SMCC = −0.49; 95% CI = [−0.95, −0.04], *p* = 0.033). Considering the meta-regression, the moderation effect of the number of sessions was significant (SMCC = −0.03; 95% CI = [−0.07, 0], *p* = 0.050), indicating that, for every additional session, the effect size SMCC was expected to decrease by 0.03. In line with previous guidelines [83], publication bias analyses were not performed due to the high between-study heterogeneity.

### 4.2. Secondary Endpoint: Depressive Symptoms

The meta-analysis was run on four effect sizes computed from four papers [98,104,105,106] (total number of participants = 37, mean age = 41.4, SD = 3.3). Here, we computed the effect of NiBS on depressive symptoms. The random-effects model highlighted a significant moderate effect of the treatment on symptom reduction (SMCC = −0.71; 95% CI = [−1.1, −0.3], *p* < 0.001; see Figure 4 for the forest plot).

The Q-statistic I^2^ suggests a low heterogeneity among studies, with Q_(3)_ = 2.9, *p* = 0.409, and 0%, while the PIs CI included zero [−1.5, 0.1]. 

## 5. Discussion 

GD is a severe health concern, with a growing prevalence worldwide. Despite the availability of a wide range of therapies, including psychological and pharmacological interventions, a gold-standard treatment effective in reducing GD’s severity and high relapse rate has not been identified yet. Moreover, the commitment of GD patients to standard care is generally low, resulting in a limited number of patients seeking treatment and high dropout rates. Innovative and effective treatment options are thus required. 

In recent decades, novel insights from animal studies and neuroimaging research provided evidence of the underlying neural mechanisms of SUDs and GD [19,20,21,22,23,51]. In the case of SUDs, the loss of control over drug use involves stable brain changes responsible for the long-lasting nature of the behavioral abnormalities. Although drug abuse comprises chemically divergent molecules, their actions in the brain activate the mesolimbic dopamine system—a primary reward system—concurrently with reducing activity in the frontostriatal system involving the prefrontal cortex [20]. Consequently, the interest in influencing maladaptive brain activity and connections has grown. One non-invasive, well-tolerated, and low-side-effects method to influence brain functioning activity is NiBS. NiBS is receiving considerable attention across multiple disorders as monotherapies or add-ons to pharmacological and psychological treatments [121,122,123,124]. 

Considering SUDs, in recent decades several attempts have been made to quantify the effectiveness of NiBS in reducing craving levels, with different meta-analyses available on the topic [67,125,126,127,128,129]. Despite promising results within SUDs, only a few works have applied NiBS to GD, thus reflecting the limited attention the disorder has received in general up to recent years [19]. Crucially, stimulation protocols were typically transferred from SUDs to GD based on the overlapping features of the two disorders. However, some discrepancies between the two conditions are also present, as highlighted in a recent review by Gomis-Vicent and colleagues [130], who unveiled cognitive and neurophysiological similarities and differences between SUDs and behavioral addictions. Compared with healthy individuals, both patient groups showed higher impulsivity, which was negatively correlated with the gray matter volume of the bilateral insula, amygdala, hippocampal complex, and parahippocampal gyri. SUDs and GD patients showed increased brain activity in the dorsomedial PFC and dorsal anterior cingulate cortex during craving induction and abnormal connectivity patterns among prefrontal regions and the amygdala. However, specific cognitive and neural patterns were found for the two conditions. GD patients typically had higher compulsivity and fewer working-memory deficits compared with SUDs. Anatomically, GD patients showed alterations in the reward network circuit, suggesting a higher sensitivity to positive reinforcement. The authors highlighted the need for a deeper understanding of cerebral features specific to GD and behavioral addictions to develop disorder-specific NiBS protocols design. Our work, among others, points in this direction.

As mentioned in the Introduction, however, only one previous systematic review focused on NiBS applied to GD as a treatment tool [71]. The authors screened the literature up to December 2019 and included 11 studies. Of these, three were case reports/series, two were open-label studies, and six were randomized sham-controlled studies. Overall, the studies included a limited number of participants. Indeed, six out of the eleven studies had fewer than ten patients. The authors highlighted an enormous heterogeneity considering the number of sessions, outcome measures, and stimulation parameters used across protocols, suggesting the need for more methodologically robust and statistically powered studies. In the present work, we aimed to update this review and, crucially, we tried to quantify the available results through a meta-analysis. Therefore, we screened English-written articles including individuals with GD diagnosis employing a NiBS technique (TMS or tDCS) to reduce craving. Overall, only ten studies satisfied our eligibility criteria, thus confirming that research on the application of NiBS to GD is still in its infancy.

A preliminary consideration comparing the included works concerns the general quality of the studies’ design and reported information. Several methodological issues, such as the small sample sizes (ranging from 4 to 30 patients), the scarcity of follow-ups, the lack of a pre-specified analysis plan, and the limited information considering the randomization process, were reflected in the risk of bias judgments, generally evaluated as “some concerns” or “fair”.

Studies were similar considering the gender of the patients in the samples, which included mainly males (N = 140 out of a total of N = 159). This point is in line with the reported gender-related differences in the epidemiological rates of GD, which seems to be more likely to occur in men than in women [7]. In all but one article [81], patients were treatment-seeking individuals. Most studies included drug-free or stable-drug-dose participants, with stabilization intervals ranging from 7 days to 6 months.

Most of the included articles, except [81,106], targeted the DLPFC (see Figure 5 for a graphical representation), in line with previous works targeting craving in SUDs [129,131,132,133] and other psychiatric disorders [43,121,134]. Indeed, DLPFC is involved in higher-order cognitive processes, such as top-down executive functions, inhibition, and attention [135,136,137].

Considering the studies’ results, most of them reported improvements in craving scores [81,98,100,102,103,104,106]. However, two studies did not find effects of NiBS on gambling urge after rTMS [105] or differences between real and sham stimulation [101]. Only a few studies assessed the effectiveness of NiBS in reducing gambling behavior, highlighting controversial results. Indeed, a reduction in the average time spent gambling was found by Pettorruso and colleagues [104], whereas another did not show the effectiveness of NiBS (Gay et al., 2017). Globally, no side effects were reported, thus providing evidence of the suitability of NiBS within the GD population. Discrepancies among the included studies were traceable considering the outcome measures used to evaluate NiBS efficacy in reducing craving, consisting of structured questionnaires or the VAS (the limitations of this approach are discussed in the next paragraph).

To quantify the efficacy of NiBS in craving reduction, we ran a meta-analysis including studies that used validated questionnaires or VAS scores to assess craving. When available, we preferred to use the former type of measure due to the well-known effect of desirability when completing the VAS [15].

Three papers were excluded from the meta-analysis [99,100,101], and only two of the seven remaining studies included a control condition [81,102]. Therefore, our analysis computed the standardized mean change in the craving scores before and after the real intervention.

The main result of the meta-analysis was a moderate but significant effect of the NiBS intervention in reducing craving. Given the limited amount of data included, we only ran a meta-regression analysis, including the number of NiBS sessions as moderator. Its effect was significant (*p* = 0.050), suggesting that increasing the number of sessions reduced craving scores, in line with results from previous studies and meta-analyses on various psychiatric and neurological conditions [128,138,139,140,141]. 

As GD patients often exhibit mood disorders [8,9,11], we ran an exploratory meta-analysis as a secondary endpoint on pre-post depressive symptoms scores measured through standardized self-report questionnaires or clinician-administered scales (BDI, HAM-D). The preliminary results highlighted that NiBS intervention decreased depressive symptoms, although a correlation with craving reduction was not found (*p* = 0.528). Interestingly, the findings on depressive symptoms were significant, despite the limited number of studies (N = 4). However, this result was somehow expected, considering the well-known effect of NiBS on depressive symptoms and, more generally, on negative emotion regulation when stimulating prefrontal regions [69,70,142,143].

Despite the potentially promising outcomes, the interpretation of the present findings requires some caution and critical discussion concerning the current state-of-the-art limitations addressed in the following paragraph.

### Limitations and Future Directions

As a first crucial point, the lack of a control condition limits the conclusion concerning the effectiveness of NiBS on GD. The low proportion of studies including a sham condition likely depends on ethical considerations preventing half of the clinical sample from receiving a potentially helpful treatment. Although the results suggest a decrease in craving scores after treatment, we cannot rule out that the observed changes were due to a placebo effect. GD patients are typically sensitive to placebo responses [19,144], although placebo or expectations effects in NiBS have also been documented in healthy individuals [145,146,147] and other clinical populations [148,149,150,151]. Previous authors hypothesized that the placebo response could be a component of the therapeutic response to NiBS [151] and—probably related to general motivation aspects—to favorable outcome treatments [152,153,154]. We believe that more randomized clinical trials are needed to disentangle whether the NiBS effect found in the present meta-analysis is trustworthy or primarily due to a placebo effect.

Another limitation of the revised literature concerns the outcome measures, which primarily addressed pre- and post-craving scores, aligning with other systematic reviews and meta-analyses investigating NiBS treatment for addiction [41,66,69,70]. Several authors highlighted that treatment outcome measures in GD are poorly defined and inconsistent across studies (see [39] for a review). Crucially, recovery lacks a shared and unequivocal definition per se, which brings us to a recursive point in which it is difficult to measure the efficacy of a treatment if we do not know how to define effectiveness and how to measure it. In line with previous authors, we believe that GD treatment should be evaluated through more ecological and long-term measures to corroborate the effects of NiBS, such as the frequency and extent of gambling behavior, relapse rate, and assessment of psychosocial or other functioning domains [155].

Indeed, concerning outcome measures, another crucial point regards the stability of the effects as most of the included studies measured NiBS efficacy only in the short term. For instance, in many cases, craving measures were limited to pre- and post-treatment time points. Only two studies measured craving at follow-ups 30 days [106] and 3 and 6 months [98] after the end of the stimulation protocol. One work [105] measured follow-up through family interviews without specifying their contents. Unfortunately, inconsistency in follow-up measures is typical when considering NiBS studies. We believe that more effort should target such a critical issue to clarify whether NiBS is an effective short-term therapy, or if the induced changes are more prolonged.

As a third point of discussion, studies differed considering individual (and brain) states at the time of NiBS delivery. Some included research provided stimulation while participants were at rest, whereas others delivered stimulation immediately after or during craving induction [98,99,100,101,102]. Moreover, Zack et al. [81] induced craving after rTMS delivery. In line with the previous literature [122,123,124], we believe that future studies should investigate treatments that time-locked NiBS with cognitive or behavioral activity. Indeed, neuroscience research highlights that NiBS effects are state-dependent, meaning that the state of the target network during stimulation influences cortical activity, excitability, and behavioral responses [156,157,158,159,160,161]. For example, when tDCS is applied at rest, current typically impacts the default-mode network’s activity, which comprises brain regions activated when individuals are not engaged in attention-demanding or goal-directed tasks [162,163,164]. Conversely, when stimulation is applied during task execution, its effects are typically traceable in the network involved in the task [156,162,165]. Moreover, the endogenous activity induced by a task has been suggested to be crucial for detecting the cathodal tDCS effect [165] compared with a resting state condition [166]. Crucially, rTMS protocols can be preferentially used as a priming (before the task) or as a consolidator (after the task) [122] due to the noise and somatosensory discomfort elicited during stimulation. Therefore, several authors addressed this point by administering cue-inducing tasks or stimuli as a primer to activate and engage the brain network and then delivering stimulation (typically through inhibitory protocols) [167,168]. Conversely, tDCS is easier to combine with concurrent treatments, as was performed in one of the works included in the present systematic review [98].

To conclude, we highlight an ultimate warning regarding the relevance of considering individual differences in response to NiBS [169,170] across studies. Both positive and null results may cover the prevalence of individuals benefiting from stimulation compared with those who do not. A deeper understanding of biomarkers and factors influencing the effectiveness of NiBS is crucial to increase knowledge of NiBS mechanisms and protocols’ efficacy, thus (hopefully) reducing heterogeneity among studies. Moreover, interindividual differences related to GD should be considered, as huge heterogeneity exists across patients suffering from the disease. Features such as the duration of the dependence, its severity, and comorbidity with other SUDs or psychiatric conditions might play a crucial role in treatment outcomes. Typically, the studies reviewed by the present work did not consider those features. A better and more exhaustive comprehension of the disease would be reached only by considering all the factors involved in gambling disorder’s complex genesis and prognosis. More research following this direction is thus encouraged.

## 6. Conclusions

The findings highlighted by this work are promising, but still limited evidence of NiBS feasibility in alleviating GD symptoms is available. Such results partially reflect the lack of specific measures to investigate GD and the absence of standardized protocols to treat it. We believe that the new classification of the disorder in DSM-5 and ICD-11 could be a starting point to prompt further research in the field, trying to solve current limitations, and potentially combining treatments to address the complexity of the disease.

## Figures and Tables

**Figure 1 brainsci-13-00698-f001:**
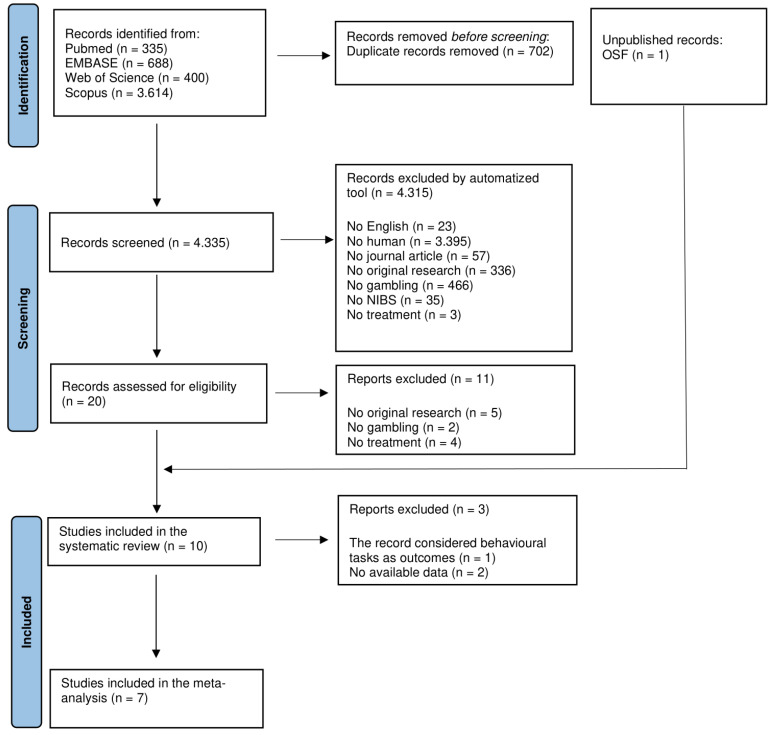
Flow chart of study selection. NiBS = non-invasive brain stimulation, OSF = Open Science Framework.

**Figure 2 brainsci-13-00698-f002:**
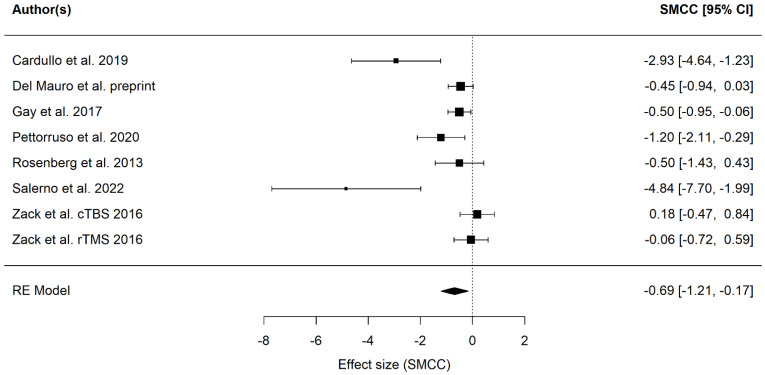
Forest plot of the effect size of NiBS on craving scores. Cardullo et al. 2019 [103], Del Mauro et al. preprint [98], Gay et al. [102], Pettorruso et al. [104], Rosenberg et al. [105], Salerno et al. [106], Zack et al. [81].

**Figure 3 brainsci-13-00698-f003:**
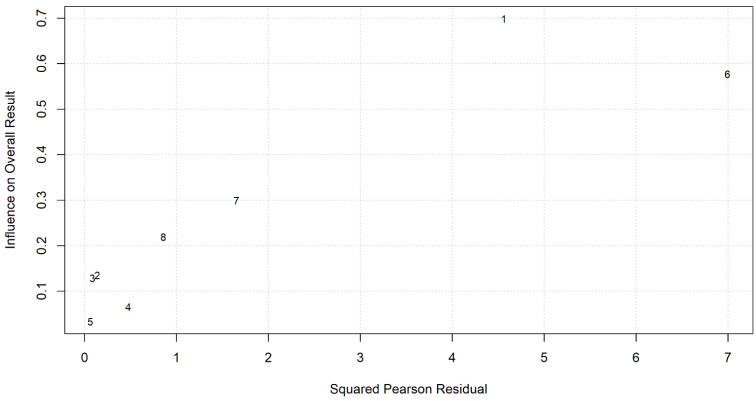
Baujat plot inspection.

**Figure 4 brainsci-13-00698-f004:**
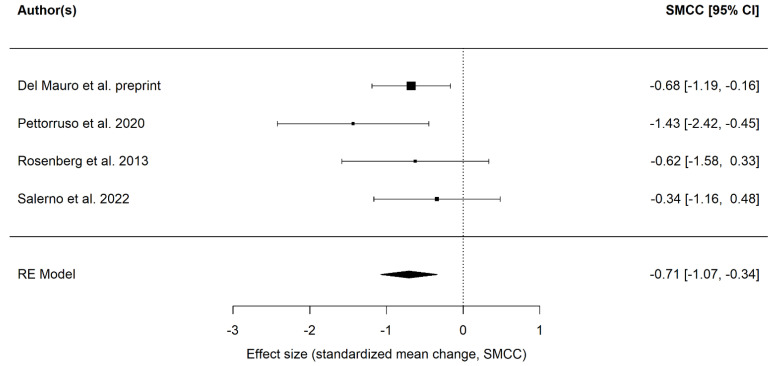
Forest plot of the effect size of NiBS on depressive symptoms. Del Mauro et al. preprint [98], Pettorruso et al. [104], Rosenberg et al. [105], Salerno et al. [106].

**Figure 5 brainsci-13-00698-f005:**
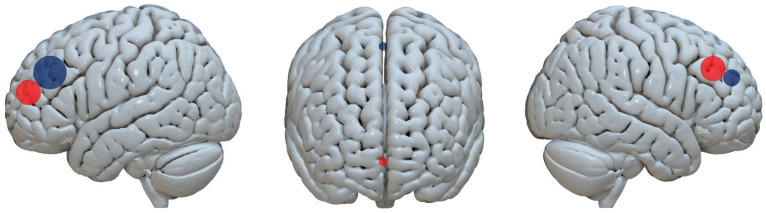
Type of stimulation and target regions of the included studies. Red dots indicate excitatory stimulation protocols (i.e., anodal tDCS and high-frequency rTMS) and blue dots indicate inhibitory stimulation (i.e., cathodal tDCS and low-frequency rTMS). The size of the dots corresponds to the number of studies that applied an excitatory or inhibitory protocol over a specific region. Brain images were modified from Surf Ice (https://github.com/neurolabusc/surf-ice) accessed on 17 March 2023.

**Table 1 brainsci-13-00698-t001:** Risk of bias assessment related to the papers evaluated through the Revised Cochrane Risk-of-Bias 2 Tool.

Cochrane Risk of Bias 2 Tool						
References	Randomization Process	Deviations from Intended Interventions	Missing Outcome Data	Outcome Measures	Reported Results Selection	Overall Judgment
Gay et al. [102]	Low	Low	Low	Low	Some concerns	Some concerns
Martinotti et al. [100]	Some concerns	Low	Low	Low	Some concerns	Some concerns
Sauvaget et al. [101]	Some concerns	Low	Low	Low	Some concerns	Some concerns
Soyata et al. [99]	Some concerns	Low	Low	Low	Some concerns	Some concerns
Zack et al. [81]	Some concerns	Low	Low	Low	Some concerns	Some concerns

**Table 2 brainsci-13-00698-t002:** Risk of bias assessment related to the papers evaluated through the NIH tool.

National Institutes of Health (NIH) Quality Assessment Tool	
References	Overall Judgment
Cardullo et al. [103]	Fair
Del Mauro et al. [98]	Good
Pettorruso et al. [104]	Good
Rosenberg et al. [105]	Poor
Salerno et al. [106]	Fair

**Table 3 brainsci-13-00698-t003:** Descriptive information of patients’ samples.

Authors	Sample	Age (Mean ± SD)	Education (Mean ± SD)	Diagnosis and Criteria	Gambling Preferences (N)	Non-Drug Interventions	Medications
Cardullo et al. [103]	7 (7 M)	42.1 ± 5.7	12 ± 3.2	CUD comorbidity with GD, DSM-5 criteria	Slot machines (6) Online poker (1)	NR	Stable pharmacological therapy throughout treatment
Del Mauro et al. [98]	18 (16 M)	41.6 ± 14.8	10.8 ± 3.3	GD, DSM-5 criteria	NR	Psychological and educational support	Nine patients were under stable therapy during treatment
Gay et al. [102]	22 (14 M)	51 ± 13.7	10.9 ± 1.4	GD, DSM-5 criteria	Slot machines (9) Scratch cards (7) Horserace betting (5) Sports betting (1)	NR	Nine patients were under stable pharmacological therapy before (1 month) and during treatment.
Martinotti et al. * [100]	4 GD/34 total sample	37.2 ± 10.4 (total sample)	NR	GD—no criteria reported	NR	NR	Twenty-two patients (total sample) were under stable pharmacotherapy
Pettorruso et al. [104]	8 (7 M)	40.6 ± 11.2	13.5 ± 3.1	GD, DSM-5 criteria	NR	Weekly psychosocial support	Four patients were under stable (6 months) pharmacological treatment.
Rosenberg et al. [105]	5 (5 M)	37.8 ± 10.3	14.8 ± 2.7	GD, DSM-IV-TR criteria	Internet gambling (1) Slot machines (3) Scratching tickets (1)	None	One patient was under stable pharmacological therapy during the treatment
Salerno et al. [106]	6 (5 M)	45.7	NR	GD, DSM-5 criteria	NR	NR	NR
Sauvaget et al. * [101]	30 (27 M)	Range: 28–56	NR	GD, DSM-IV criteria	Eight participants usually gambled online	NR	Stable (at least 7 days before treatment)
Soyata et al. * [99]	20 (20 M)	37.2 ± 10.3	13.4 ± 3.2	GD, DSM-5 criteria	NR	NR	NR
Zack et al. [81]	9 (9 M)	43.2 ± 13.2	NR	PG, DSM-IV criteria	NR	NR	No medication allowed

Alcohol-use disorder (AUD); cocaine-use disorder (CUD); Diagnostic and Statistical Manual (DSM); gambling disorder (GD); male (M); pathological gambling (PG); not reported (NR); substance-use disorder (SUD). * Studies included in systematic review but not in the meta-analysis.

## Data Availability

Data sharing not applicable.

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
