# Peer review of "Betting on Non-Invasive Brain Stimulation to Treat Gambling Disorder: A Systematic Review and Meta-Analysis"

_brainsci, 2023, doi:10.3390/brainsci13040698_

Round 1

Reviewer 1 Report

Abstract

Please report the main results of the meta-analysis in the abstract (effect size and p-value).

No conclusion was included in the abstract. Besides, the background of the research was too long in the abstract. The motivation and objective of the study should be clearly stated in the abstract.

Introduction:

It is not clear whether internet gambling disorder is also included in the review.

Please cite relevant reviews on the topic of NIBS on substance addiction and gambling disorders.

The neural mechanism of NIBS in revising gambling-related brain pathology should be added.  

Results:

The descriptive information that has been mentioned in the table can be removed from the results part.

Figure 2: The effect size of Salermo et al was huge. Please check the data and perform sensitivity analyses.

In meta-analysis, the authors need to state that the effect size is drawn from the comparison between theverum and sham stimulation.

The authors can perform explanatory meta-regression and subgroup analyses.

Discussion,

There is no need to repeat the results in the discussion.

Reviewer 2 Report

The topic is interesting, however , the following comments can improve the quality of the paper.

The paper only lists previous studies and lacks analysis and comparison of results.

Introduction section: please explain why this study is important, motivation.

list the limitations of the state of the art, and explain author's suggestion to tackle those limitations.

The paper contains some informal language, please check the writing and improve paper presentation. 

Round 2

Reviewer 2 Report

My comments have been addressed.